# Nutritional status and associated factors among hospitalized adult patients in comprehensive specialized hospitals of Western Ethiopia

Birtukan Kebede Moti[1]*, Gudina Egata[2], Desalegn Wirtu Tesso[1]

1 Department of Public Health, Institute of Health Sciences, Wollega University, Nekemte, Ethiopia,
2 Department of Nutrition & Dietetics, School of Public Health, College of Health Sciences, Addis Ababa University, Addis Ababa, Ethiopia

* birtu2020phd@gmail.com

## Abstract

### Background

Malnutrition is common among hospitalized patients but is often overlooked in low- and middle-income countries like Ethiopia. Therefore, this study aims to determine its prevalence and identify associated factors among adult patients admitted to comprehensive specialized hospitals in Western Ethiopia.

### Methods

A facility-based cross-sectional study was conducted among 1,106 randomly selected adult patients admitted to comprehensive specialized hospitals from December 16, 2023 to April 16, 2024. The sample was proportionally allocated by patient population. Data were collected using Kobo Toolbox, then exported from Microsoft Excel to Stata version 17 for analysis. Nutritional status was assessed at admission using the Subjective Global Assessment (SGA). Ordinal logistic regression was applied after verifying the proportional odds assumption, with the Brant test used for model validation. Adjusted odds ratios (AORs) alongside 95% confidence intervals (CIs) were computed, and statistical significance was set at p < 0.05.

### Results

Among all study participants, 24.5% (95% CI: 21.8%, 27.4%) were severely malnourished, while 34.4% (95% CI: 31.4%, 37.6%) were moderately malnourished. In the multivariable ordinal logistic regression analyses, factors significantly associated with both moderate and severe malnutrition included age greater or equal to 65 years [(AOR = 4.32; 95% CI: (2.02, 9.28)], age 41–64 years [(AOR = 3.36; (1.58, 7.16)], male sex [(AOR = 1.46, 95% CI:(1.085, 1.953)], urban residence [(AOR = 0.49; 95% CI: (0.32, 0.76)], lack of formal education [(AOR = 5.37; 95% CI: (3.32, 8.69)],

**Data availability statement:** All relevant data are within the paper and its Supporting information files.

**Funding:** The author(s) received no specific funding for this work.

**Competing interests:** The authors declare that they have no financial or non-financial competing interests.

comorbidities at admission [(AOR = 1.50;95%CI:(1.04,2.16)], and inadequate dietary diversity [(AOR = 1.37;95% CI:1.01,1.86)].

## Conclusion

Malnutrition among hospitalized adult patients remains high and is strongly associated with age, sex, urban residence, educational status, comorbidities at admission, and dietary diversity. Therefore, hospital-based nutritional interventions should adopt a multidisciplinary approach, including early screening and identification, proper management of comorbidities, dietary counseling, and the development of comprehensive care plans and guidelines to improve clinical outcomes.

## Introduction

The European Society for Clinical Nutrition and Metabolism (ESPEN) defines malnutrition -specifically undernutrition as "a state resulting from lack of intake or uptake of nutrition that leads to altered body composition, notably decreased fat free mass and body cell mass, resulting in diminished physical and mental function and impaired clinical outcomes from disease" [1]. In 2022, an estimated 390 million adults aged 18 years and older worldwide were underweight [[2].

Globally, an estimated 20–50% of patients are either malnourished or at high risk of malnutrition at the time of hospital admission [3]. Evidence from developed countries indicates substantial prevalence rates among hospitalized adults, including 46.3% in the United States, 29.6% in China, 24.1% in Germany, 46.5% in Ankara, Turkey, 43.0% in northern Delaware, and 51% in Canada [4–9], as well as 48.7% in Argentina [10] and 57.3% in Tehran [11]. In contrast, available evidence from Africa suggests an even higher burden. For example, the prevalence of malnutrition was reported to be 60% in Zambia [12]. In Ethiopia, studies have documented similarly high magnitudes, with 55.1% in referral hospitals of the Amhara National Regional State [13], 64.32% in southern Ethiopia [14], and 62.1% at Tikur Anbessa Specialized Hospital [15].

Malnutrition is a common and serious problem in the hospital setting, yet it frequently goes undetected among newly admitted inpatients. Adults are at increased risk of developing malnutrition due to multifactorial causes, including co-morbidities and their complications, such as polypharmacy, inflammation and pain [9,16]. In addition, factors such as underlying illness, socio-economic status, lack of knowledge, certain diagnostic or therapeutic procedures, lack of standardized nutrition care, lack of monitoring of nutritional status, and other conditions affecting food intake contribute to the occurrence of malnutrition among hospitalized adults [9,15]. Moreover, variables such as education level, employment, urbanization, food insecurity, income level, and age also play a significant role as predictors of malnutrition [9,13,17,18].

The developmental, economic, social and medical impacts of the global burden of malnutrition are profound and long lasting, affecting individuals, families, communities, and nations alike. Malnutrition is not only a consequence of disease but can

also accelerate disease progression or even act as a causative factor. Among hospitalized patients, malnutrition has been associated with an increased risk of infectious and non-infectious complications, poor wound healing, muscle wasting, hospital readmission, delay in rehabilitation, treatment failure, prolonged hospital stays, higher mortality, and increased health care costs. Despite these serious consequences, malnutrition in hospitalized adults is often unrecognized, underreported, and left untreated [2,3,5,19].

Conversely, the nutritional status of hospitalized adult patients in Ethiopia is often overlooked, with greater emphasis placed on vulnerable groups such as children under five years of age and pregnant and lactating women. Malnutrition remains a significant public health concern in the country, affecting both the general population and hospital patients. However, limited attention has been given to investigating malnutrition among hospitalized adults, with only a few studies conducted in selected regions of Ethiopia [13–15]. In particular, the magnitude of malnutrition among adult inpatients has not been adequately explored in hospitals in western Ethiopia. Overall, evidence on adult malnutrition within healthcare settings remains scarce. Therefore, this study aimed to determine the prevalence of malnutrition and its associated factors among adult patients admitted to comprehensive specialized hospitals in western Ethiopia.

## Materials and Methods

### Study setting, design, and period

A facility based cross-sectional study was conducted in comprehensive specialized hospitals located in the East Wollega zone of the Oromia Regional State in western Ethiopia from December 16, 2023 to April 16, 2024. Wollega University Comprehensive Specialized Hospital has a total of 230 beds, while Nekemte Comprehensive Specialized Hospital has 203 beds. Both hospitals were equipped with weight and height measuring scales in each ward; however, only Wollega University Specialized Hospital had a dedicated nutrition specialist assigned to each unit. According to reports from the Oromia Regional Health bureau, Zonal Health bureau, and the Health Information Management System of the respective hospitals, these facilities provide health care services to more than 3.1 million people in the East Wollega zone, 1.5 million people in West Shewa, and 1.2 million people in the Buno Bedelle zone [20,21].

### Study participants, and eligibility criteria

Adult patients aged ≥18 years admitted to medical, surgical and orthopedic wards in comprehensive specialized hospitals were included in the study. However, patients with physical disabilities such as kyphosis, scoliosis, and limb deformities, as well as pregnant women, unconscious or comatose patients, surgical emergency cases and those expected to be discharged within 72 hours were excluded from the study.

### Sample Size determination and Sampling Procedure

A single population proportion formula was used to calculate the sample size based on the stated assumptions, resulting in a final sample size of 1106.

$$n = \frac{\left(Z_{\left(\frac{\alpha}{2}\right)}\right)^2 P(1-P)}{d^2}$$

Where n denotes the sample size, $Z_{\alpha/2}$ represents the standard normal value corresponding to a 95% confidence level (±1.96), *P* is the estimated proportion of malnutrition from a previous study (62.1%) [15] d is the margin of error, and a 10% non-response rate was considered.

Wollega University Comprehensive Specialized Hospital and Nekemte Comprehensive Specialized Hospital were purposively selected based on the types of services they provide. Medical, orthopedic, and surgical wards were included

according to the eligibility criteria. The total sample size was proportionally allocated to each hospital based on the number of admissions in the selected wards over one quarter without disaggregation on a daily and monthly basis, ensuring adequate representation of each ward. A simple random sampling technique was used to select study participants from the admission registration records each morning. All eligible patients admitted within the preceding 48 hours were identified and assigned unique identification numbers. Participants were then selected using a computer -generated random number method.

## Data Collection tools, procedures, and outcome measurement

Trained B.Sc. nurses with experience in data collection, clinical practice, and proficiency in local languages were recruited to collect the data. Socio-demographic information—including age, sex, residence, living arrangements, and education status-was obtained. In addition, data on diagnosis and the presence of comorbidities were collected using a pretested structured questionnaire.

Patients' nutritional status was assessed within 48 hours of admission using a validated tool, the Subjective Global Assessment (SGA). The SGA includes both medical history and physical examination components. The medical assessment covers five domains: (i) recent weight change or loss, (ii) dietary intake, (iii) gastrointestinal symptoms, (iv) functional status, and (v) underlying disease conditions. The physical examination component focuses on identifying clinical signs of malnutrition, such as loss of subcutaneous fat, muscle wasting, edema, and ascites [22,23].

Percentage weight loss (PWL) was calculated using the standard formula based on usual and current body weight: self-reported usual body weight (pre-illness) in kgs minus current body weight in kgs divided by the usual body weight, and multiplied by 100. Current body weight was measured using a calibrated portable electronic scale (Seca) with an accuracy of 100 g. Patients were weighed wearing light clothing and no shoes, with heavy items such as wallets and mobile phones removed. Dietary intake was assessed based on reported reductions in food consumption over a specified period and was considered significant when patients consumed less than 50% of the meals provided. Gastrointestinal symptoms such as nausea, vomiting, dysphagia, and diarrhea were also evaluated as part of the nutritional assessment. Increased metabolic demand, particularly in inflammatory conditions such as infections, contributes to malnutrition through enhanced catabolism and muscle breakdown. Functional status was assessed using a Takei Physical Strength Dynamometer. Participants were instructed to apply maximum grip strength with their dominant hand and maintain the contraction for a few seconds. Measurements were taken in a standing position, with shoulders back, feet evenly placed at hip-width apart, the elbow fully extended, and the arm held neutrally without touching the body. For patients unable to stand, measurements were conducted in a seated position [24]. Additional assessments included evaluation of subcutaneous fat loss in the lumbar, upper arm, orbital, and thoracic regions. Muscle wasting was assessed by examining the quadriceps and deltoid muscles. The presence of edema was also evaluated in the ankles, sacral area, and abdomen to identify signs of fluid accumulation, including ascites.

## Operational definitions

Nutritional status: The seven-point SGA scoring system was used to classify patients into three categories of nutritional status: well nourished (scores 6–7), mildly to moderately malnourished (scores 3–5), and severely malnourished (scores 1–2) [22].

The results were interpreted according to the established SGA criteria [22,23] as follows:

Well nourished (SGA-A): no reduction in food or nutrient intake, <5% weight loss, no or minimal symptoms affecting intake, no functional impairment, and no evidence of fat or muscle mass depletion.

Mildly to moderately malnourished (SGA-B): definite reduction in food or nutrient intake, 5–10% weight loss without stabilization or regain, mild to moderate symptoms affecting intake, moderate functional impairment or recent decline, and mild to moderate loss of fat and/or muscle mass.

Severely malnourished (SGA-C): severe reduction in food or nutrient intake, >10% ongoing weight loss, marked symptoms affecting intake, and severe functional impairment, and pronounced loss of fat and/or muscle mass.

Dietary diversity: was assessed using the minimum dietary diversity for women (MDD-W) approach, with consumption of at least five or more of ten food groups in the previous 24 hours considered adequate. The food groups included: (1) starchy staples (grains, white roots, tubers, and plantains), (2) legumes (beans, peas, and lentils), (3) nuts and seeds, (4) dairy products, (5) flesh foods (meat, poultry, and fish), (6) eggs, (7) dark green leafy vegetables, (8) vitamin A-rich fruits and vegetables, (9) other vegetables, and (10) other fruits [25].

Presence of comorbidity: In this study comorbidity denotes the presence of common health problems observed among the patients during admission such as congestive heart failure, diabetes mellitus, hypertension, tuberculosis, HIV/AIDS, and urinary tract infection among others which can have an association with nutritional status of the patients.

Semi-urban: In Ethiopia, it is not a strictly standardized administrative category (like *urban* or *rural* defined by the government), but it is commonly used in research and planning to describe areas that are in transition between rural and urban settings [26].

## Data quality management

Data collectors and supervisors received comprehensive training on the study objectives, data collection tools, ethical considerations, and data recording procedures. A pre-test was conducted on 5% of the sample size at Gimbi Hospital to ensure the reliability and clarity of the instruments. Based on the findings, necessary modifications were made before the commencement of the actual data collection. Data collection in each unit/ward was conducted by a single trained professional to minimize bias and ensure consistent interpretation of the SGA. Data completeness was checked daily by supervisors, while the principal investigator continuously monitored the overall quality of the data throughout the data collection period.

## Statistical analysis

The data were downloaded from Kobo Toolbox in Microsoft Excel format, cleaned, and exported to Stata version 17 for analysis. Nutritional status was coded "0' for well-nourished patients (SGA-A), representing the lowest risk or reference category; "1" for moderately malnourished patients (SGA-B); and "2" for severely malnourished patients (SGA-C), indicating the most severe condition. Descriptive statistics, including frequencies, means, standard deviations, and proportions, were computed for selected socio-demographic variables.

An ordinal logistic regression model was used to identify predictors of malnutrition among adult patients. Multicollinearity was assessed using the variance inflation factor (VIF), with values below 1.014, indicating no evidence of multicollinearity among the independent variables. Model fitness was evaluated using the Pearson and Deviance goodness-of-fit tests. The model fitting information showed a statistically significant result ($p = 0.001$). The proportional odds (parallel lines) assumption of the ordinal logistic regression model was tested using the Brant test, and a non-significant p-value confirmed that the assumption was satisfied. Adjusted odds ratios (AORs) with their corresponding 95% confidence intervals (CIs) were calculated to assess the strength of the association between dependent and independent variables. Statistical significance was declared at a P value of less than 0.05.

## Ethical considerations

Ethical approval was obtained from the Research Ethics Review Committee (RERC) of Wollega University (Ref no: WU/Ro/716). The study was conducted in accordance with the relevant guidelines and regulations of the Declaration of Helsinki. Accordingly, participants' right to voluntary participation, confidentiality, and protection from harm was fully respected. Written informed consent was obtained from all study participants after clearly explaining the objectives of the study prior to data collection. In addition, all personal identifiers, including names and other identifying information, were kept confidential and anonymous throughout the study.

## Results

### Socio-demographic Characteristics

A total of 927 adult hospitalized patients participated in the study, yielding an 84% response rate. More than half of the participants, 484 (52.2%), were admitted to medical wards, while 382(41.2%) were from surgical wards. Of the total respondents, 486(52.4%) were male. The mean (±SD) age was 39.37 (±16.1) years. Approximately one quarter, 228 (24.6%), were unable to read and write. The majority of participants, 489 (52.8%), were Protestant Christians, and 669 (72.2%) were married. Regarding occupation, 336 (36.2%) were farmers, and slightly more than half, 472 (50.9%), lived in urban areas (Table 1).

### Nutritional status of hospitalized patients

Based on the SGA assessment, out of the 927 hospitalized adult patients who participated in this study, 24.5%, 95% CI: (21.8%, 27.4%) were severely malnourished, while 34.4%, 95% CI: (31.4%, 37.6%) were moderately malnourished. In

**Table 1. Socio-demographic characteristics of hospitalized adult patients in comprehensive specialized hospitals of Western Ethiopia, 2025.**

| Characteristics | | Frequency | Percent |
|---|---|---|---|
| **Sex** | Male | 486 | 52.4 |
| | Female | 441 | 47.6 |
| **Age** | 18-40 | 548 | 59.1 |
| | 41- 64 | 306 | 33.0 |
| | ≥65 | 73 | 7.9 |
| **Educational status** | Have no formal education | 228 | 24.6 |
| | Primary | 238 | 25.7 |
| | Secondary | 157 | 16.9 |
| | high school | 145 | 15.6 |
| | College/University | 159 | 17.2 |
| **Religion** | Muslim | 111 | 12 |
| | Orthodox | 315 | 34 |
| | Protestant | 489 | 52.8 |
| | Catholic | 4 | 0.4 |
| | Others** | 8 | 0.9 |
| **Occupation** | Civil Servant | 80 | 8.6 |
| | Farmer | 336 | 36.2 |
| | Merchant | 70 | 7.6 |
| | Housewife | 384 | 41.4 |
| | Private employ | 48 | 5.2 |
| | Others* | 9 | 1.0 |
| **Marital Status** | Unmarried/Single | 202 | 21.8 |
| | Married | 669 | 72.2 |
| | Widowed | 40 | 4.3 |
| | Divorced | 11 | 1.2 |
| | Separated | 5 | 0.5 |
| **Residence** | Urban | 472 | 50.9 |
| | Rural | 318 | 34.3 |
| | Semi urban | 137 | 14.8 |

** = Waqefta, those don't have religion, *= daily laborers, housekeeping.

addition, 41.1%, 95% CI: (37.9%, 44.3%) of the participants were well-nourished. Overall, the magnitude of malnutrition was [58.9%, 95%CI: (55.5%−62.0%)] (Fig. 1).

## Distribution of malnutrition by socio-demographic variables

In this study, participants aged 65 and above had the highest burden of malnutrition, with 39.7% being moderately malnourished and 47.9% severely malnourished. Among male participants, 26.1% were severely malnourished. Regarding educational status, 39.4% of those who were unable to read and write were moderately malnourished, while 37.2% were severely malnourished. In addition, 43.1% of participants living in semi-urban areas were moderately malnourished. Among patients with a history of previous admissions, 36.5% had moderate malnutrition and 29% had severe malnutrition. With respect to clinical characteristics, 31.1% of patients admitted due to gastrointestinal diseases were severely malnourished. Among those, with co-morbidities, 25.4% were severely malnourished and 40.5% were moderately malnourished. Furthermore, Furthermore, 36.1% of participants with inadequate dietary diversity were moderately malnourished, while 29% were severely malnourished (Table 2).

## Factors associated with malnutrition among hospitalized adult patients

In multivariable ordinal logistic regression analysis, it was found that being 65 years and above, male sex urban residence, lack of formal education, presence of co-morbidities upon admission, and inadequate dietary diversity at the household level were significantly associated with moderate and severe malnutrition.

When all other variables were held constant, hospitalized adult patients aged 65 years and above had 4.32 times [(AOR = 4.32; 95% CI: (2.02, 9.28)] higher odds of being malnourished (moderate or severe) compared with those aged 18–40. Similarly, patients aged 41–64 years had approximately 3.4 times [(AOR = 3.36; (1.58, 7.16)] higher odds of malnutrition compared to their counterparts. On the other hand, male patients had 1.46 times [(AOR = 1.46; 95% CI: 1.085, 1.953) higher odds of being moderately and severely malnutrition compared to female patients. In addition, patients living in urban areas had 51% [(AOR = 0.49; 95% CI: 0.32, 0.76)] lower odds of experiencing moderate or severe malnutrition compared to those living in semi-urban areas. Adult patients with no formal education had 5.37 times [(AOR = 5.37; 95% CI: (3.32, 8.69)] higher odds of being moderately or severely malnourished compared to those with a college or university level education. Similarly, patients with Primary education had approximately 3.2 times [(AOR = 3.16, (2.03, 4.91)] higher odds of malnutrition. Regarding comorbidities, patients who had comorbid conditions at admission were 1.5 times [(AOR = 1.5; 95% CI: 1.04, 2.16)] more likely to be moderately or severely malnourished compared to those without

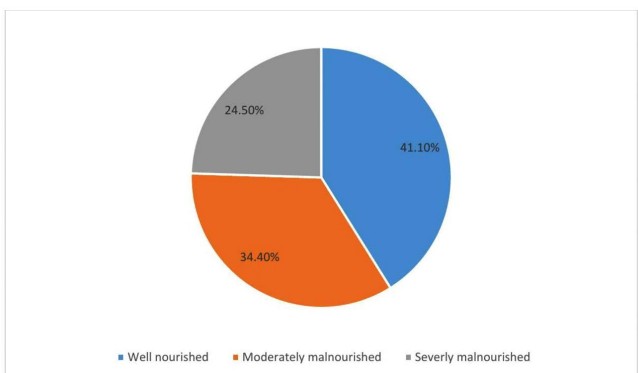

**Fig 1. Nutritional status of hospitalized adult patients in comprehensive specialized hospitals of Western Ethiopia, 2025.**

**Table 2. Distribution of malnutrition by socio-demographic variables among hospitalized adult patients, Western Ethiopia, 2025 (n = 927).**

| Variables | | Well-nourished (SGA: A) N = 382 | Moderate malnourished (SGA:B) N = 318 | Sever Malnourished (SGA:C) N = 227 |
|---|---|---|---|---|
| Age | 18 - 40 | 234(42.7) | 188(34.3) | 126(23) |
| | 41 - 64 | 139(45.4) | 101(33.0 | 66(21.6) |
| | ≥65 | 9(12.3) | 29(39.7) | 35(47.9) |
| Sex | Male | 194(39.9) | 165(34) | 127(26.1) |
| | Female | 188(42.6) | 153(34.7) | 100(22.7) |
| Residence | Urban | 201(42.6) | 154(32.6) | 117(24.8) |
| | Rural | 136(42.8) | 105(33) | 77(24.2) |
| | Semi Urban | 45(32.8) | 59(43.1) | 33(24.1) |
| Educational status | Have no formal education | 54(23.4) | 91(39.4) | 86(37.2) |
| | Primary | 82(34.7) | 93(39.4) | 61(25.80 |
| | Secondary/high school | 150(49.7) | 89(29.5) | 63(20.90 |
| | Collage/University | 96(60.8) | 45(28.5) | 17(10.8) |
| Primary diagnosis at Admission | Cardiovascular | 38(39.2) | 35(36.1) | 24(24.7)) |
| | Gastrointestinal | 79(35.6) | 74(33.3) | 69(31.1) |
| | Genitourinary | 29(41.4) | 22(31.40 | 19(27.1) |
| | Respiratory | 50(42.0) | 40(33.6) | 29(24.4) |
| | Neurology | 15(50.0) | 11(36.7) | 4(13.3) |
| | Musculoskeletal | 58(39.5) | 55(37.4) | 34(23.1) |
| | Hematology | 41(41.8) | 36(36.7) | 21(21.1) |
| | Others# | 72(50) | 45(31.1) | 27(18.8) |
| History of previous admission | Yes | 88(34.5) | 93(36.5) | 74(29) |
| | No | 294(43.8) | 225(33.5) | 153(22.8) |
| Co-morbidity | Yes | 63(34.1) | 75(40.5) | 47(25.4) |
| | No | 382(41.2) | 243((32.7) | 180(24.3) |
| Dietary Diversity practice | Adequate | 293(43.6) | 226(33.6) | 153(22.8) |
| | Inadequate | 89(34.9) | 92(36.1) | 74(29) |
| Infection at admission | Yes | 56(31.8) | 78(44.3) | 42(23.9) |
| | No | 326(43.4) | 240(32) | 185(24.6) |

#= other admission diagnosis

comorbidities. Moreover, individuals with inadequate dietary diversity at the household level had 1.37 times [(AOR = 1.37; 95%CI: (1.01, 1.86)] higher odds of malnutrition compared to their counterparts (Table 3).

## Discussion

This study aimed to assess the magnitude of malnutrition and its predictors among adult patients admitted to comprehensive specialized hospitals of Western Ethiopia. The findings showed that 58.9% of the patients were malnourished, of whom 24.5% were severely malnourished and 34.4% were moderately malnourished. Malnutrition was significantly associated with being aged 65 years and above, male sex, urban residence, lack of formal education (inability to read and write), presence of co-morbidities at admission, and inadequate dietary diversity at household level.

In this study, the overall magnitude of malnutrition among adult hospitalized patients was 58.9%. This magnitude is higher than reports from China (29.6%), Germany (24.1%), Ankara (46.5%), Canada (51%), and northern Delaware

**Table 3. Results of multivariable ordinal Logistic Regression analysis of factors associated with malnutrition among hospitalized adult patients, Western Ethiopia, 2025 (n = 927).**

| Variable | | β coefficient | Standard Error | Z | P>Z | AOR(95% CI) |
|---|---|---|---|---|---|---|
| Age | 41–64 | 1.212 | 0.386 | 9.87 | 0.002 | 3.36(1.58, 7.16)* |
| | ≥65 | 1.464 | 0.390 | 14.12 | 0.0001 | 4.32(2.015,9.28)** |
| Sex | Male | 0.376 | 0.150 | 6.28 | 0.012 | 1.46(1.085,1.95)* |
| Residence | Urban | 0.718 | 0.223 | 10.34 | 0.001 | 0.49(0.325,0.76)* |
| | Rural | 0.342 | 0.233 | 2.16 | 0.142 | 0.71(0.45,1.12) |
| Marital status | Married | 0.203 | 0.108 | 3.51 | 0.061 | 0.82(0.66, 1.009) |
| | Others# | | | | | |
| Educational status | Have no formal education | 1.681 | 0.246 | 46.90 | 0.0001 | 5.37(3.32,8.69)** |
| | Primary education | 1.149 | 0.225 | 26.10 | 0.0001 | 3.16(2.03,4.91)** |
| Occupation | Daily Laborers | 0.074 | 0.039 | 3.63 | 0.057 | 1.08 (0.99,1.16) |
| | Others## | | | | | |
| History of previous admission | Yes | 0.226 | 0.177 | 1.63 | 0.202 | 0.79(0.56,1.13) |
| Co-morbidity | Yes | 0.405 | 0.185 | 4.77 | 0.029 | 1.49(1.04,2.16)* |
| Dietary diversity | Inadequate dietary diversity | 0.315 | 0.156 | 4.07 | 0.044 | 1.37(1.009,1.86)* |
| Cut 1 | | 0.128 | 0.155 | | | 0.128(−0.176, 0.431) |
| Cut 2 | | 1.630 | 0.165 | | | 1.630(1.307, 1.953) |

*= P value <0.05 and **= P value ≤0.001; Adjusted OR: odd ratio. #=Divorced & Separated, ##=Civil Servant, Farmer, Merchant, and Private employ

(43.0%) [4–9], all of which also used the Subjective Global Assessment (SGA) method. This discrepancy may be explained by geographical variations, including differences in food security status, as well as variations in socio-demographic and socioeconomic characteristics of the study populations, disease burden, and disparities in health system capacity and medical infrastructure. The relatively high magnitude also indicates that malnutrition is not limited to traditionally vulnerable groups such as children and pregnant or breastfeeding women, but is also highly prevalent among hospitalized adult patients, thereby warranting increased attention. Studies conducted in Latin America have reported a prevalence ranging from 40–60% at hospital admission [19], and Tehran also reported a prevalence of 57.3% [11], which is comparable to the findings of the present study. This similarity may be attributed to the use of comparable nutritional assessment methods across studies. Similarly, several previous studies conducted in different parts of Ethiopia reported comparable findings, including 42.2% at hospitals of Amhara National regional state [13], 64.32% at Wolaita Sodo University Comprehensive Specialized Hospital in Southern Ethiopia [14], and 62.1% at Tikur Anbessa Specialized Hospital [15]. Nevertheless, the magnitude observed in the current study is higher than that reported in studies from Iran (23.92%) [27], Korea (22.0%) [28], and a university hospital in Bobo Dioulasso, Burkina Faso (32.4%) [29].

Malnutrition was found to be associated with male sex in this study, where male hospitalized adult patients had 1.46 times higher odds of experiencing moderate or severe categories of malnutrition compared to their female counter parts, This finding is consistent with studies conducted in Argentina [10], Iran [27], Brazil [30], and South Africa [31]. This difference may be explained by the higher energy requirements of males, which may not be adequately met during illness compared to females.

The odds of being classified as moderately or severely malnourished were 4.32 times higher among patients aged 65 years and above compared to hospitalized adults aged 18–40 years. Similarly, those aged 41–64 years had nearly 3.4 times higher odds of moderate or severe malnutrition relative to the reference age group. Comparable findings have been reported in studies from Argentina [10], Tehran [11], Korea [28], and Brazil [30]. This suggests that older adults are particularly vulnerable to malnutrition due to physiological changes such as reduced appetite, the presence of comorbidities, and functional decline, as well as psychological and socio-economic factors that further increase their risk.

Patients with no formal education were found to have higher odds of malnutrition compared to their educated counterparts in this study. This finding is consistent with studies conducted in Tehran [11] and Ethiopia [13]. This may be explained by the fact that education enhances individuals' awareness of health conditions, improves their understanding of nutritional information, and equips them with the skills needed to make appropriate dietary choices.

This study indicates that urban residence is associated with malnutrition among hospitalized patients. However, to date, there is limited evidence specifically examining the relationship between place of residence and malnutrition, assessed using SGA in hospitalized populations, highlighting the need for further research to better understand this association among adult inpatients. Although studies from Korea, Brazil, and Ethiopia have assessed the prevalence and outcomes of malnutrition using SGA, they have not considered residence as a predictor [13,28,30]. The higher odds of malnutrition observed in urban areas may be explained by less diversified dietary intake among adults living in urban and semi-urban settings, potentially driven by urban poverty, unhealthy eating patterns, and a higher burden of chronic diseases.

The study found that the presence of comorbidities at admission was a significant risk factor for malnutrition. This finding was consistent with studies conducted in Argentina [10], Brazil [30], Ethiopia [13] and a systematic review from Latin America [19]. The observed association may be explained by the fact that comorbid conditions can impair immune function, delay recovery, compromise treatment outcomes, and increase susceptibility to malnutrition. Therefore, patients with comorbidities require early identification and prompt management to reduce the risk of nutritional deterioration. This suggests that insufficient dietary intake may compromise immune function and increase the risk of malnutrition, particularly among hospitalized patients.

In this study, inadequate dietary diversity was also strongly associated with malnutrition among hospitalized patients, which is consistent with a previous study conducted in Ethiopia reporting that low dietary diversity is linked to malnutrition [13]. This suggests that insufficient dietary intake may compromise function and increase the risk of malnutrition, particularly among hospitalized patients.

This study could have the following limitations. First, the use of a single assessment method without alternative or complementary approaches may have limited a more comprehensive evaluation of nutritional status. Although participants were given clear instructions on how to recall their usual weight, reliance on self-reported weight may have introduced information bias due to measurement error, potentially leading to under or overestimation of the proportion of malnutrition. In addition, early hospital discharge or patient transfer, as well as refusal to participate due to fatigue or lack of interest, may have contributed to a reduced response rate and potential non-response bias, which could have affected the estimated proportion of malnutrition in this study.

## Conclusions

The magnitude of malnutrition is high among adult hospitalized patients. Malnutrition was significantly associated with patients' age, sex, educational status, presence of comorbidities at admission, and dietary diversity. Therefore, hospital-based nutritional interventions should prioritize early identification of malnutrition, effective control of comorbidities, counseling on diversified diet. In addition, the development and implementation of standardized hospital nutrition care plans and guidelines are essential to improve patient outcomes. Furthermore, further studies are recommended to strengthen nutritional policies and inform interventions in both hospital and community settings in Ethiopia.

## Supporting information

**S1 File. Questionnaires.**
(DOCX)

**S2 File. Data.**
(SAV)

## Acknowledgments

We would like to express our sincere gratitude to Wollega University for its comprehensive support throughout the study. We are also grateful to the data collectors, head nurses, and hospital administrators for their valuable assistance in facilitating the data collection process. Finally, we deeply appreciate all study participants for their willingness to take part in the study.

## Author contributions

**Conceptualization:** Birtukan Kebede Moti, Gudina Egata, Desalegn Wirtu Tesso.

**Data curation:** Birtukan Kebede Moti, Gudina Egata, Desalegn Wirtu Tesso.

**Formal analysis:** Birtukan Kebede Moti, Gudina Egata, Desalegn Wirtu Tesso.

**Funding acquisition:** Gudina Egata, Desalegn Wirtu Tesso.

**Investigation:** Birtukan Kebede Moti, Gudina Egata, Desalegn Wirtu Tesso.

**Methodology:** Birtukan Kebede Moti, Gudina Egata, Desalegn Wirtu Tesso.

**Project administration:** Birtukan Kebede Moti, Gudina Egata, Desalegn Wirtu Tesso.

**Resources:** Birtukan Kebede Moti, Desalegn Wirtu Tesso.

**Software:** Birtukan Kebede Moti, Gudina Egata.

**Supervision:** Birtukan Kebede Moti, Gudina Egata, Desalegn Wirtu Tesso.

**Validation:** Birtukan Kebede Moti, Gudina Egata, Desalegn Wirtu Tesso.

**Visualization:** Birtukan Kebede Moti, Gudina Egata, Desalegn Wirtu Tesso.

**Writing – original draft:** Birtukan Kebede Moti, Gudina Egata, Desalegn Wirtu Tesso.

**Writing – review & editing:** Birtukan Kebede Moti, Gudina Egata, Desalegn Wirtu Tesso.

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
