## [Decision Letter · Decision Letter 0]

23 Jan 2026

PONE-D-25-57131Nutritional status of hospitalized adult patients in comprehensive specialized hospitals of Western, Ethiopia.PLOS One

Dear Dr. Moti,

Thank you for submitting your manuscript to PLOS ONE. After careful consideration, We found the topic of your manuscript to be highly relevant and timely, addressing a significant gap in the current literature. The potential contribution of your work is considerable. However, the reviewers have raised several substantive concerns that must be adequately addressed before the manuscript can be considered for acceptance. Therefore, we invite you to submit a revised version of the manuscript.

We look forward to receiving your revised manuscript.

Kind regards,

Nagasa Dida, MPH

Academic Editor

PLOS One

Journal Requirements:

I have read the journal policy and no authors of this manuscript have competing interests.

5. Please amend the manuscript submission data (via Edit Submission) to include author Gudina Egata

6. Please amend your authorship list in your manuscript file to include author Gudina Egata Atomsa

Additional Editor Comments:

The superscript 1 and 3 on the first and last authors are affiliated with the same institution; hence, use the same number.The latest version of Stata is 19, which was released in April 2025. Is there a Stata version 23?In the abstract section you have to indicate the data collection method. You have used the Subjective Global Assessment tool, which assesses the patient's nutritional status by using patient medical history and the physical examination. Hence, it is imperative to indicate how you approached and when the data collection was held, whether at admission, a few periods after admission, or on discharge.Compressive specialized hospitals—who were these hospitals? In the sampling procedure you haven’t indicated how many specialized comprehensive hospitals there are and how many of them were considered.The patients might be admitted to the hospital with other medical problems other than malnutrition, and the clinicians might overlook the malnutrition of the patients. Plus, in addition to this, the patients may stay a long time at the hospital based on the nature of the disease they are admitted for. During his/her stay, the patient might become malnourished because of the hospital-related services. From the ethical point of view, is there anything done to resolve their malnutrition, and have you checked their nutritional status on discharge? If there was a case that malnourished patients were discharged, how would you see it from the ethical point of view?

Reviewers' comments:

Reviewer's Responses to Questions

**Comments to the Author**

1. Is the manuscript technically sound, and do the data support the conclusions?

Reviewer #1: Partly

Reviewer #2: Yes

Reviewer #3: Yes

Reviewer #4: Partly

Reviewer #5: Yes

Reviewer #6: Yes

Reviewer #7: Yes

Reviewer #8: Partly

2. Has the statistical analysis been performed appropriately and rigorously?

Reviewer #1: No

Reviewer #2: I Don't Know

Reviewer #3: Yes

Reviewer #4: No

Reviewer #5: Yes

Reviewer #6: Yes

Reviewer #7: Yes

Reviewer #8: Yes

3. Have the authors made all data underlying the findings in their manuscript fully available?

Reviewer #1: Yes

Reviewer #2: Yes

Reviewer #3: Yes

Reviewer #4: Yes

Reviewer #5: Yes

Reviewer #6: Yes

Reviewer #7: Yes

Reviewer #8: Yes

4. Is the manuscript presented in an intelligible fashion and written in standard English?

Reviewer #1: No

Reviewer #2: No

Reviewer #3: Yes

Reviewer #4: No

Reviewer #5: Yes

Reviewer #6: Yes

Reviewer #7: Yes

Reviewer #8: Yes

5. Review Comments to the Author

Reviewer #1: Manuscript Title:

Nutritional status of hospitalized adult patients in comprehensive specialized hospitals of Western, Ethiopia

Reviewer Name

Samuel Abate Walda (BSc, MSc in adult health nursing, lecturer)

Overall Recommendation:-Major Revision

1. Summary of the Study

The authors conducted a facility-based cross-sectional study assessing the prevalence of malnutrition and associated factors among 927 adult hospitalized patients in two comprehensive specialized hospitals in Western Ethiopia. Nutritional status was evaluated using the Subjective Global Assessment (SGA). The study reports a high burden of malnutrition (58.9%) and identifies several predictors, including age, sex, education, comorbidities, and dietary diversity.

2. General Assessment

The topic is important, timely, and relevant to global hospital nutrition and public health in low-resource settings. The manuscript is generally well organized, the methods are mostly sound, and the findings are potentially valuable for clinicians and policymakers.

However, major methodological clarifications and corrections are needed before the paper can meet PLOS ONE standards for scientific rigor. Several issues exist regarding sample size justification, statistical coding errors, and variable definitions, clarity of SGA assessment procedures, inconsistent reporting, and figure/table formatting.

With substantial revision, the manuscript has the potential for publication.

3. Major Concerns

1. Sample size mismatch (critical)

The authors report an intended sample size of 1,106, yet only 927 respondents were included (84% response rate).

However, they do not discuss:

Why the response rate was low.

How non-response may affect representativeness.

Whether power was recalculated or compromised.

Action required:

Provide a justification for the final achieved sample and discuss implications of non-response.

2. Coding error in the statistical analysis section

The manuscript states:

“coded ‘0’ for well-nourished… ‘1’ for moderately malnourished… and ‘2’ indicating middle category, and ‘2’ for severely malnourished.”

There is a clear duplicated value (‘2’) for two different categories, which is a serious methodological problem.

Action required:

Correct the coding scheme and confirm that model outputs are based on correctly ordered categories.

3. Use of SGA requires more detail to ensure reproducibility

The SGA assessment process is insufficiently described. Examples of missing details include:

Who performed SGA? Were they trained nutritionists or only nurses?

Was inter-rater reliability assessed?

Was SGA validated in similar Ethiopian hospital settings?

How were subjective components standardized?

Action required:

Provide clear description of SGA training, standardization, quality control, and inter-observer agreement.

4. Dietary diversity measurement inconsistencies

Two different descriptions of dietary diversity appear:

One uses 11–12 food groups (including coffee/tea, sugar, oils, etc.).

Another uses the FAO MDD-W 10-food group standard.

This inconsistency undermines the validity of the dietary diversity variable.

Action required:

Clarify which dietary diversity tool was used, ensure alignment with valid guidelines, and re-analyze if needed.

5. Interpretation of AOR for residence appears reversed

The result states:

“Urban residence AOR = 0.49… lower odds of experiencing moderate or severe malnutrition compared to semi-urban”

But earlier, the authors claim:

“Urban residence was associated with undernutrition”

This is contradictory.

Action required:

Clarify reference category, interpretation, and correct contradictory statements.

6. Confounder selection justification is missing

PLOS ONE requires transparency in model building. The authors do not describe:

How variables were selected for the multivariable model (p < 0.25? theoretical framework?)

Whether multicollinearity was assessed.

Whether any variables were excluded due to collinearity or small cell sizes.

Action required:

Explain confounder selection strategy and report tests for multicollinearity.

7. Ethical issues regarding consent

The statement “thumbprint written consent” is ambiguous. Clarification needed:-

Was the thumbprint used on a written consent form?

Who served as witness?

Were illiterate participants informed through a standardized script?

8. Literature review has inaccuracies or outdated references

The section is overly long and contains inaccuracies:

Some percentages do not match cited papers.

Narrative occasionally repeats findings unnecessarily.

Referencing style is inconsistent with PLOS ONE format.

Action required:

Revise the introduction for conciseness, accuracy, and updated references.

4. Minor Comments

Abstract

Grammar and flow need slight refinement.

Methods should explicitly mention analysis software once (avoid repetition).

Tables

Table 1: "Secondary" and "Secondary/High school" categories appear redundant.

Table 2: Some parentheses are mismatched, and percentages do not always sum logically.

Table 3: Occupational categories unclear; “others” is too vague.

Figures

Figure 1 is referenced but not provided in the document

Writing and clarity

Several grammatical inconsistencies require language editing.

Discussion occasionally restates results instead of interpreting them.

Referencing

Should follow PLOS ONE numeric style consistently.

DOIs should be included where available.

5. Strengths of the Manuscript

Large sample size compared to most studies in Ethiopia.

Important clinical topic with public health significance.

Use of SGA (a validated tool).

Appropriate use of ordinal logistic regression (assuming corrections are made).

Clear presentation of ethical approvals and data availability.

6. Specific Questions for Authors

How “inadequate dietary diversity” was operationally defined?

Were any patients excluded due to inability to complete SGA reliably?

Were acute infection markers (e.g., fever) considered confounders in the model?

Did the study consider hospital food services or meal frequency?

Could selection bias exist since emergency patients were excluded?

7. Recommendation to the Editor

Major Revision

The manuscript addresses a relevant public health issue and contains potentially important findings, but substantial methodological clarifications and corrections are needed to ensure scientific rigor and reproducibility. Once these major issues are addressed, the manuscript may be suitable for reconsideration.

Reviewer #2: Dear Author/s,

Thank you for conducting research on the nutritional status of hospitalized adult patients in comprehensive specialized hospitals of Western Ethiopia. It gives a good overview of the nutritional status of hospitalized adult patients in western Ethiopia. In your research paper I noticed the following points, which I am not clear on.

1.In the Abstract section, on line 40. You wrote “predictors terminology,” but in the results section you wrote association factors. Please use consistent terminology.

2.Also, your sample size is inconsistent. 1106 patients participated in the study in the abstract section, but 927 patients participated in the result section. How could it be a different figure?

3.On line 64, what is hospital malnutrition? That means patients developed undernutrition due to their hospitalization? It is better if replaced by malnutrition among adult patients admitted to the hospitals.

4.I am not satisfied with the conclusion section! Only you rewrote your results in the conclusion section. It is better if you add the consequence or impact of undernutrition.

5.On line 90, the first sentence of the paragraph needs a citation.

6.Line 110, regarding the gap of the study, I am not satisfied. Studies that were conducted in Ethiopia have limitations? In terms of variable or study design? Or did you because they are few in number? What were their gaps?

7.line 125, you stated, “According to the Oromia Regional Health Office, Zonal Health Office, and Health Information Management System of each hospital, these hospitals provide health care services for more than 3.1 million people in the East Wollega zone, 1.5 million people in West Shewa, and 1.2 million people in the Buno Bedele zone.” Do you mean that each hospital has its own separate catchment population? Is it possible? Please clarify!

8.Line 130, gynecology ward: was it included or excluded?

9.line 132 you stated, “…..However, patients with physical disabilities such as kyphosis, scoliosis, and limb defects including deformities, as well as pregnant patients, unconscious or comatose patients, obstetrics patients, surgical emergency patients, and patients to be discharged within 72 hours were excluded from the study.” Is there a difference between pregnant and obstetric patients? Why did you mention obstetrics patients separately?

10.If you excluded patients who discharged within 3 (72 hrs.) days, how could you get this sample size? What was the hospital’s average length of hospital stay?

11.In lines 154, 155, and 156, the assessment was made by whom? Nurse or physician, but you didn't include physician in the data collection procedure. So justify your data quality?

12.Have you excluded patients who have no lower extremities (amputated patients) in the orthopedics ward?

13.Line 193 you have did pretest. But you didn’t state the hospital. At the same hospital?

14.Line 301 Please add CI

15.line 318 in the discussion section, you stated, “Nevertheless, the results of the current study is found to be lower than results of a few studies reported from Iran (23.92%) (20), Korea (22.0%) (21), and the university hospital in Bobo-Dioulasso, Burkina Faso (32.4%) (22).” It needs justification

16.Line 365, in the conclusion section. Do you think only hospital-based intervention could solve the malnutrition problem? I think this undernutrition prevalence was not due to hospitalization. Why didn’t you recommend community-based intervention?

17.It is better if you add recommendations like strengthening nutritional policy at Ethiopian hospitals.

Reviewer #3: Dear reviewer, thank you for inviting me for reviewing the paper. I congratulate the authors for excellent job. It is written paper, and the are is also very wonderful. I have forwarded few comments that researchers can include for the better of the work.

Study title: Nutritional status of hospitalized adult patients in comprehensive specialized hospitals of Western, Ethiopia.

Comment: The author comes with excellent area which is timely and seeks immediate action. However, the title and the study objective don’t inline. I wonder if the author re-writes the title as “Nutritional status and associated factors of hospitalized adult patients in comprehensive specialized hospitals of Western, Ethiopia”

1.1. Background

Comment: The study is cross-sectional study and the last sentence of abstract should have to avoid ‘predictor’ because it better fit for cohort study and it is difficult address predictor using cross-sectional study design.

1.2. Method

How was the proportional Samples being proportionally allocation to each hospital based on their patient population size was made? Was it ward based? Some period admission? or what else. It has to identified.

1.3. Conclusion

I hope the first sentence of this sub-section is to mean. “Hospitalized patient malnutrition….”

It will be good if the sub-section also involves the recommendation.

Discussion of is well written. However, I wonder if the researcher add implication for each result like: for practice, policies, and future research.

What are the strength and limitation of current study so that readers can use it effectively.

What are the key recommendation of the researchers from current finding?

I wonder if researchers add declaration section for the manuscript as per the journal guideline.

Reviewer #4: HELLO DEAR AUTHORS

i have reviewd your manuscript, consider the following comments for improving

first

you aims to assess the magnitude of malnutrition and its predictors

among adult patients admitted to comprehensive specialized hospitals in Western,

Ethiopia" as your target populatiuon were the admitted patients, you have to provide clear justification regarding your target settings? othewise you need to assess the hospital aquired malnutrtion (malnutrition arised after admission) which is different of community malnutrition !

you have to use term prevelance not magnitude as a quantatitive not qulaitiative study

second:

you stated that you have selected the sample by simple random technique, but you didnt explain how randomization were ensured (random number, Random lottery, list of population...

third: tool

-pretested structured questionnaire and recorded line 151 ?? it not mentioned before nor explaine

-measurmnts are subjetive tools ( self reported wight !!

- how you assesssd the subcatenoues fat and muscle waste ??

it is crucial to amandes the provide major comment (aim, population, and tools& measures ) then editing, language and improving result reported could be emphasized

Reviewer #5: Reviewer Comments

Introduction

Line 105: A citation is required to support this claim.

Line 112: The introduction states the aim is to examine “predictors of malnutrition,” but the concept is broad. Please specify which predictors are examined in this study (e.g., sociodemographic, clinical, dietary, functional) and provide a brief rationale for why these predictors were selected and why they matter in the study context.

Methods

Line 150: Please clarify how comorbidity was assessed (e.g., clinical diagnosis from records, self-report, Charlson Comorbidity Index, number/type of chronic conditions). If a tool/index was used, provide a reference and scoring approach.

Lines 152–155: The manuscript states that a validated SGA was used, but no reference is provided. Please cite the original SGA reference and/or any validation work relevant to the study population/setting.

Lines 165–167: Please provide a reference for the handgrip strength measurement procedure. Also, clarify the number of trials (one vs multiple),

Lines 175–177: This paragraph requires revision for clarity. Please explain how dietary intake is captured within the SGA (i.e., what intake domains were assessed, how “reduced intake” was defined, and the reference timeframe used).

Line 185: Please justify the choice of the dietary diversity tool. Why was dietary diversity selected rather than other dietary assessment approaches (e.g., 24-hour dietary recall, food frequency questionnaire)? If dietary diversity was chosen due to feasibility, cost, respondent burden, or validation in similar settings, please state this explicitly and cite supporting references.

Discussion

Line 306: The manuscript suggests malnutrition may shorten hospital stay. Please clarify whether length of stay was measured/analyzed in this study.

Lines 305–309: The rationale for comparing your findings with studies from Tehran and the USA is unclear. Please explain why these settings are appropriate comparators (e.g., similar case mix, assessment method, healthcare context) and interpret similarities/differences in a way that is meaningful for the reader.

Line 319: The statement about men requiring more calories than women should be supported with an appropriate reference (e.g., dietary reference intakes/energy requirement guidelines).

Lines 322–328: A key finding is that adults aged ≥65 years are at higher risk of malnutrition. The discussion in this paragraph should focus more deeply on this result:

Why are older adults at higher risk in this setting?

Which physiological changes are relevant (and cite evidence)?

Are there contextual factors (e.g., comorbidity, functional decline, appetite changes, socioeconomic factors)?

The current section spends too much space on cross-study prevalence comparisons without adequately interpreting this study’s main finding.

Lines 339–341: Please provide a citation to support this claim.

Lines 342–347: The discussion on comorbidity and malnutrition is difficult to interpret because comorbidity measurement is not clearly described. Please clarify what comorbidity variables represent in your analysis and how they should be interpreted.

Lines 348–354: “Inadequate dietary diversity” needs definition. Please specify:

what threshold defined “inadequate,” and

what this means nutritionally in the context of the population studied.

Line 354: If this is presented as a limitation, please explain why it is a limitation?

Line 355: Please clarify what is meant by “impact of hospitalization.” Is this referring to appetite, restricted diets, disease severity, fasting for procedures, or length of stay?

Line 356: Please state what could be used instead in future studies (e.g., 24-hour recall, repeated recalls, plate waste measures, direct observation), considering feasibility in hospital settings.

Line 357: Please describe if there are any steps taken to reduce bias.

Limitations

Please acknowledge that self-reported usual body weight may introduce measurement error and could over- or underestimate malnutrition prevalence. If any strategies were used to minimize this, please report them.

Conclusion

The conclusion currently reads largely as a summary. Please tailor recommendations more directly to the key findings.

Reviewer #6: General comments

The authors address one of the most critical public health priorities by focusing on adults—the most economically productive yet often overlooked age group. This study has the potential to illuminate future interventions and inform targeted responses to the emerging challenges identified. The findings are particularly alarming and warrant serious attention from relevant stakeholders. Overall, the paper is informative and explanatory; however, substantial revisions are required before it can proceed to the next stage. The authors are strongly encouraged to carefully review the specific comments below and revise the manuscript thoroughly, line by line.

Specific comments

Title:

Nutritional status of hospitalized adult patients in comprehensive specialized hospitals of Western Ethiopia

Cover page:

Affiliations indicated by superscript 1 and 3 seems similar but written twice. Please check it

Abstract

Line 54: Results

Introduction

Lines 82-86: The authors state that ample evidence exists demonstrating the burden of malnutrition; however, it remains unclear which specific forms of malnutrition and which settings are being referenced. Clarification is also needed on whether the study directly addresses the gap identified in lines 87–88. Furthermore, the gaps outlined on lines 87–88 would be more compelling if supported with additional contextual details. The three Ethiopian studies reviewed (references 13, 14, and 15) should be explicitly compared to highlight similarities and differences in setting, population, and outcomes, and to clearly demonstrate how the present study advances knowledge beyond existing evidence.

The introduction in general needs re-organization. It would clearly depict for the readers if organized in the way that what is the problem to how much it is disastrous and why? The gap identified should have been indicated clearly.

Methods

Line 162/3: This sentence should be recaptured. “Current was measured using 162 an electronic portable scale (Seca scale) with an accuracy of 100 grams.”

Lines 160-161: Self-reported data were used to calculate the PWL. However, 24.6% of the study participants had no formal education, which may have inevitably affected the accuracy of self-reporting. Moreover, this variable is described as a key predictor of malnutrition, yet it is unclear how the authors arrived at this conclusion. A detailed explanation is needed on how the potential bias and effect modification related to educational status were assessed and managed in the analysis.

Line 182: The explanatory variables are not to the contrary to the response variables. Please modify the phrase. For example, on the other hand,…..

Line 211: The p-value, confirming the proportional odds assumptions, which the authors checked should be indicated

Results

Table 1: The educational status of the participants is listed twice as “secondary,” which requires clarification or correction. The distinction between educational categories is important, as misclassification may influence the study outcomes.

In the same table, the category “other” under religion is marked with a dollar sign, which conveys unintended meaning. The authors should replace this with a neutral symbol and ensure that the symbol is formatted appropriately as a superscript or subscript.

Table 2: The education status in this table is categorized into 4; but totaling the both secondary category in the first table. The authors are requested to recheck this controversy before proceeding.

Table 3: The two categories of educational status (secondary, twice) now missed from the table. Which category is the confirmatory (reference)? The controversy with tables continued. As this variable is an important for your study, this should be solved before proceeding. You are requested to rerun your model after correcting it.

Table 3: Occupation===others&==the symbol you used needs to be changed. The previous comment helps. How the authors categorize and analyze is confusing. It either be well described in the analysis section of the method parts.

Discussion

Line 302,303: The variation in the magnitude of malnutrition is primarily attributed to geographical differences and other potential explanations—such as disparities in healthcare access, socio-demographic and economic conditions, and differences in study population size. However, the scientific rationale for emphasizing geographic factors is not clearly articulated, but needed. The discussion should provide evidence-based justification for the role of geographical setting and convincingly explain why it is prioritized over alternative explanations. The authors are strongly encouraged to substantiate all proposed explanations in the discussion section with appropriate scientific reasoning and supporting evidence.

Line 322-324: This sentence is a vague sentence. Would you reconsider. Your interpretation is expected to use simple sentence. Consider the audience.

Line 333: The association between the residence and malnutrition is not well described. The final table and line 276-278, indicated that living in urban areas is protective while your explanation in this discussion section is not satisfactory. This needs clarity and even more elaboration on the link between the residence and malnutrition. This is the hot public health agenda of the time.

Lines 354-358: The limitations of the study would be more informative if discussed in greater detail. Specifically, the authors should clarify what measures were taken to minimize the stated limitations and explain how these constraints inform future research directions and opportunities for investigators in this field.

Furthermore, given that malnutrition is the primary outcome of interest, several key variables that are typically expected appear to be missing. The manuscript does not address anthropometric measurements (at minimum BMI, MUAC, or waist circumference), micronutrient status (such as anemia or deficiencies of key minerals), behavioral factors, or the economic status of respondents (at least a wealth index). The authors are expected to explicitly acknowledge the absence of these variables and discuss their implications for the interpretation of the findings. The issue of external validity should also be explained

Line 360: How high is high? This conclusion would be more persuasive if compared with established magnitude of either national or international level.

Reviewer #7: Manuscript Review Report

Thank you for this insightful manuscript. This is a well-conducted and important study addressing a relevant Nutritional related health problem particularly in Ethiopia. The writing is generally clear and structured, and the data appear robust. However, a few sections require refinement for clarity, conciseness, and improved scientific rigors. Below is my section-by-section feedback to help you strengthen the manuscript.

1. Under ethics statement section, there is a statement written as “An informed written consent was obtained from all study participants after explaining the objectives of the study prior to data collection. And from individuals who were unable to read or write, with support from their families (legal guardians).” This statement needs clarification as patients did not fill the questionnaire and I don’t see the importance of including the statement ‘who were unable to read or write, with support from their families’.

2. It is important to include the schematic presentation of the sampling procedure

3. Needs conceptual framework as it is the foundation for your study

4. While you considered associated factors, the sample size was calculated only for nutritional status. Could you clarify why the sample size was not determined for the associated factors as well?

5. In sampling procedure part you wrote the statement “Enrollment of study participants was continued until the desired sample size was reached.” I don’t think this is scientific

6. A pre-test was conducted on 5% of the sample size. Where? It should be included!

7. The results section is well structured; however, some refinements are recommended. Specifically, it would be clearer and more consistent if the socio-demographic characteristics were presented in percentages, as was done in other parts of the results

8. The conclusion section requires revision. It should provide clear directions for future practice, research, or policy

9. The manuscript currently lacks a section addressing strengths and limitations. Including this section would enhance the overall rigor of the paper by acknowledging both the contributions and the potential constraints of the study.

Reviewer #8: General comments

This manuscript addresses an important and under-researched public health problem and aligns well with PLOS ONE’s scope. The study benefits from a relatively large sample size, use of a validated assessment tool (SGA), and appropriate statistical modelling (ordinal logistic regression). However, the manuscript requires substantial revision to improve clarity, internal consistency, methodological transparency, and language quality. Several inconsistencies in sample size, variable definitions. The primary focus of the study should also be clarified. The followings are important areas to be improved by the authors

Specific Comments

Title

I suggest modifying the title to: “Nutritional status and its associated factors among adult patients admitted to …” as the authors mentioned the results of associated factors in the results, discussions and conclusion parts.

Abstract

Background

The background and objective should be consistent. Currently, the background seems to focus on malnutrition among the general adult population rather than adult patients admitted to hospitals. It would be better to align the background with the actual study population (i.e., adult admitted patients).

Methods

The authors should clearly describe the characteristics of the admitted patients. Patients with different health conditions may have different nutritional statuses; therefore, specifying these characteristics is important.

Results

The authors report a prevalence of 24.5%, but it is unclear how this figure was classified as high or low. The result should be interpreted with reference to appropriate national or international standards or guidelines.

Conclusion

Some of the recommendations are not supported by the study findings. For example, the authors recommend improving hospital nutritional care plans and guidelines; however, no results indicate the absence or inadequacy of such plans. Similarly, the recommendation that “hospital-based nutrition interventions should take into account multidisciplinary approaches” raises questions, as it is unclear whether comprehensive specialized hospitals in western Ethiopia are currently providing such services, and no findings appear to address this issue.

Introduction

Although the authors focus on malnutrition, the introduction primarily discusses under nutrition and underweight. It would be important to discuss the broader categories of malnutrition, including under nutrition, overweight, obesity, protein-energy malnutrition (PEM), and other forms, and then clearly indicate which type(s) are most prevalent among the study population.

Methods

• The authors did not report the size of the target population.

• It is important to specify the types of adult admitted patients included in the study (e.g., medical vs. surgical wards, acute vs. chronic conditions), as nutritional status may differ across these groups.

• The proportion value used for sample size calculation should be clearly stated.

• There is inconsistency in the sampling description. The authors state that simple random sampling was used, yet enrolment continued until the desired sample size was reached. This raises concerns about whether the method was truly probability-based or closer to consecutive or quota sampling.

• The sampling frame used for simple random sampling should be clearly described.

Ethical Considerations

• On page 10, line 225, and the authors’ state: “All necessary interventions were made to treat undernourished patients.”

This statement is unclear. Does it imply that the study included an intervention component? If not, clarification is needed.

Results

• Please ensure consistency in terminology. Terms such as “severely malnourished,” “severe malnutrition,” “sever malnourished,” “moderate malnutrition,” and “moderately malnourished” are used in this document interchangeably, which is confusing.

• The focus of the study should be clarified: is it malnutrition or under nutrition? The background and results sections appear inconsistent in this regard. Operational definitions for the main outcome variables and key terms should be provided.

• Table 2 is titled “Proportion of under nutrition,” yet the table content refers to malnutrition. These terms should not be used interchangeably without clarification.

• Since malnutrition includes both under nutrition and over nutrition, the authors should be specific and focused on the intended outcome.

• On page 15, lines 275–288, the focus shifts entirely to under nutrition, despite earlier references to malnutrition. This inconsistency requires careful revision.

• Table 3 should be clearer. The outcome variable should be explicitly shown, and results should be presented using appropriate cross-tabulations, including variables, COR, AOR, and p-values.

Discussion

Similar to the concerns I rose in the background, methods, and results sections, the authors also used malnutrition and under nutrition under discussion interchangeably. The discussion should clearly and consistently focus on the defined outcome variable.

Conclusion

The conclusion appears to focus primarily on under nutrition rather than malnutrition. The authors should clearly state the actual focus of the study and avoid using these terms interchangeably.

6. PLOS authors have the option to publish the peer review history of their article (what does this mean?). If published, this will include your full peer review and any attached files.

Reviewer #1: **Yes:** Samuel Abate Walda

Reviewer #2: No

Reviewer #3: **Yes:** Getahun Fetensa

Reviewer #4: No

Reviewer #5: No

Reviewer #6: No

Reviewer #7: **Yes:** Firezer Keno Belay

Reviewer #8: No

---

## [Author Response · Author response to Decision Letter 1]

8 Mar 2026

Response to Reviewers

Academic editor Comments:

Point by point response to Editor’s comments

Academic editor Comments:

• The superscript 1 and 3 on the first and last authors are affiliated with the same institution; hence, use the same number.

Response: Dear editor, thank you for pointing this out. It has been corrected in the revised manuscript.

• The latest version of Stata is 19, which was released in April 2025. Is there a Stata version 23?

Response: Dear editor, thank you for pointing this out. It was a typographical error and has been corrected in the revised manuscript.

• In the abstract section you have to indicate the data collection method. You have used the Subjective Global Assessment tool, which assesses the patient's nutritional status by using patient medical history and the physical examination. Hence, it is imperative to indicate how you approached and when the data collection was held, whether at admission, a few periods after admission, or on discharge.

Response: Dear editor, thank you for comment. It was mentioned in the methods section of the main text that the nutritional status of patients' was assessed within 48 hours of admission. However, this information was not included in the abstract section due to word limit constraints.

• Compressive specialized hospitals—who were these hospitals? In the sampling procedure you haven’t indicated how many specialized comprehensive hospitals there are and how many of them were considered.

Response: Dear editor thank you for your comment. There are two comprehensive specialized hospitals: Wollega University Comprehensive Specialized Hospital and Nekemte Comprehensive Specialized Hospital. Both of them have been included.

• The patients might be admitted to the hospital with other medical problems other than malnutrition, and the clinicians might overlook the malnutrition of the patients. Plus, in addition to this, the patients may stay a long time at the hospital based on the nature of the disease they are admitted for. During his/her stay, the patient might become malnourished because of the hospital-related services. From the ethical point of view, is there anything done to resolve their malnutrition, and have you checked their nutritional status on discharge? If there was a case that malnourished patients were discharged, how would you see it from the ethical point of view?

Responses: Dear editor thank you for your constructive comments and recommendations. To address malnutrition in patient overlooked by clinicians, as well as those who become malnourished due to hospital-related services, we report to the head nurses, clinicians and managers for necessary management.

Reviewer Comments:

Reviewer 1

However, major methodological clarifications and corrections are needed before the paper can meet PLOS ONE standards for scientific rigor. Several issues exist regarding sample size justification, statistical coding errors, and variable definitions, clarity of SGA assessment procedures, inconsistent reporting, and figure/table formatting.

With substantial revision, the manuscript has the potential for publication.

3. Major Concerns

1. Sample size mismatch (critical)

The authors report an intended sample size of 1,106, yet only 927 respondents were included (84% response rate).

However, they do not discuss:

Why the response rate was low.

How non-response may affect representativeness.

Whether power was recalculated or compromised.

Action required:

Provide a justification for the final achieved sample and discuss implications of non-response.

Responses: Thank you dear reviewer. Although the final achieved sample size was lower than the calculated 1106, the 84% response rate is acceptable and sufficient for analysis. Non-response may slightly reduce power and introduce potential bias, but the impact is limited and we have transparently reported it as a study limitation.

2. Coding error in the statistical analysis section

The manuscript states:

“coded ‘0’ for well-nourished… ‘1’ for moderately malnourished… and ‘2’ indicating middle category, and ‘2’ for severely malnourished.”

There is a clear duplicated value (‘2’) for two different categories, which is a serious methodological problem.

Action required:

Correct the coding scheme and confirm that model outputs are based on correctly ordered categories.

Responses: Thank you for pointing out the typographical error. To clarify, there are three categories in the study: “0” for well-nourished, “1” for moderately malnourished, and “2” for severely malnourished.

3. Use of SGA requires more detail to ensure reproducibility

The SGA assessment process is insufficiently described. Examples of missing details include:

Who performed SGA? Were they trained nutritionists or only nurses?

Was inter-rater reliability assessed?

Was SGA validated in similar Ethiopian hospital settings?

How were subjective components standardized?

Action required:

Provide clear description of SGA training, standardization, quality control, and inter-observer agreement.

Responses: Dear reviewer thank you for the constructive questions. All the questions raise are addressed on the documents as indicated below:

• Data collection was conducted by BSc nurses with prior experience in data collection (page 148).

• take training on the tool (SGA) (from page 193-194)

• To avoid interrater reliability only one data collector was assigned in one ward/unit( page 196-197).

• SGA was standardized based on the scores given for each medical history and physical examination components, found on revised manuscript from line154-193.

4. Dietary diversity measurement inconsistencies

Two different descriptions of dietary diversity appear:

One uses 11–12 food groups (including coffee/tea, sugar, oils, etc.).

Another uses the FAO MDD-W 10-food group standard.

This inconsistency undermines the validity of the dietary diversity variable.

Action required:

Clarify which dietary diversity tool was used, ensure alignment with valid guidelines, and re-analyze if needed.

Responses: Thank you for pointing out the typing error in the writing. FAO MDD-W 10-food group standard.

5. Interpretation of AOR for residence appears reversed

The result states:

“Urban residence AOR = 0.49… lower odds of experiencing moderate or severe malnutrition compared to semi-urban”

But earlier, the authors claim:

“Urban residence was associated with undernutrition”

This is contradictory.

Action required:

Clarify reference category, interpretation, and correct contradictory statements.

Responses: Dear Reviewer, thank you very much for your constructive insight. The error was typographical, and we have made the necessary modifications based on your comment. The changes are reflected in the revised version of the manuscript.

6. Confounder selection justification is missing

PLOS ONE requires transparency in model building. The authors do not describe:

How variables were selected for the multivariable model (p < 0.25? theoretical framework?)

Whether multicollinearity was assessed.

Whether any variables were excluded due to collinearity or small cell sizes.

Action required:

Explain confounder selection strategy and report tests for multicollinearity.

Responses: Dear Reviewer, thank you very much for your constructive comments. Information related to multicollinearity can be found on page 8, lines 219–220 of the revised manuscript.

7. Ethical issues regarding consent

The statement “thumbprint written consent” is ambiguous. Clarification needed:-

Was the thumbprint used on a written consent form?

Who served as witness?

Were illiterate participants informed through a standardized script?

Responses: Dear reviewer, yes, illiterate participants were informed through a standardized script. Thumb print is the signature used for those Illiterate who unable to put other type of signature (for those use thumb print as a signature).

8. Literature review has inaccuracies or outdated references

The section is overly long and contains inaccuracies:

Some percentages do not match cited papers.

Narrative occasionally repeats findings unnecessarily.

Referencing style is inconsistent with PLOS ONE format.

Action required:

Revise the introduction for conciseness, accuracy, and updated references.

Responses: Dear Reviewer, thank you for your constructive comment. The correction has been made.

4. Minor Comments

Abstract

Grammar and flow need slight refinement.

Responses: Dear Reviewer, thank you for your constructive comment. The correction has been made.

Methods should explicitly mention analysis software once (avoid repetition).

Tables

Table 1: "Secondary" and "Secondary/High school" categories appear redundant.

Responses: We sincerely thank the Reviewer for identifying the typographical error. The correction has been made in the Methods section and in Table 1.

Table 2: Some parentheses are mismatched, and percentages do not always sum logically.

Table 3: Occupational categories unclear; “others” is too vague.

Responses: Dear Reviewer, thank you for your comment. In Table 3, the occupational category "Others" includes individuals who work as civil servants, farmers, merchants, and private employees.

Figures

Figure 1 is referenced but not provided in the document

Responses: Dear Reviewer, thank you for your comment. Figure 1 is included in the submitted manuscript.

Writing and clarity

Several grammatical inconsistencies require language editing.

Discussion occasionally restates results instead of interpreting them.

Referencing

Should follow PLOS ONE numeric style consistently.

DOIs should be included where available.

6. Specific Questions for Authors

How “inadequate dietary diversity” was operationally defined?

Response: Dear Reviewer, thank you for your comment. Yes, lines 186–187 address the classification of adequate and inadequate dietary diversity in the manuscript.

Were any patients excluded due to inability to complete SGA reliably?

Were acute infection markers (e.g., fever) considered confounders in the model?

Response: Dear Reviewer, thank you for your comment. Yes, some patients were excluded due to inability to complete the Subjective Global Assessment (SGA) reliably. The SGA requires comprehensive medical history taking and physical examination; therefore, patients who were unable to provide adequate information and for whom reliable assessment could not be performed were excluded from the study.

Did the study consider hospital food services or meal frequency?

Response: Dear Reviewer, thank you for your question. No, this study did not specifically assess hospital food services or meal frequency.

Could selection bias exist since emergency patients were excluded?

Response: Dear Reviewer, thank you for your insightful comment. Emergency patients were excluded because their clinical instability did not allow for comprehensive nutritional assessment at admission.

Responses: Dear reviewer thank you for your constructive comments, recommendations, and questions:

Reviewer #2: Dear Author/s,

Thank you for conducting research on the nutritional status of hospitalized adult patients in comprehensive specialized hospitals of Western Ethiopia. It gives a good overview of the nutritional status of hospitalized adult patients in western Ethiopia. In your research paper I noticed the following points, which I am not clear on.

1. In the Abstract section, on line 40. You wrote “predictors terminology,” but in the results section you wrote association factors. Please use consistent terminology.

Responses: Dear Reviewer, thank you for your constructive comment. We appreciate your valuable feedback.

2. Also, your sample size is inconsistent. 1106 patients participated in the study in the abstract section, but 927 patients participated in the result section. How could it be a different figure?

Responses: Dear reviewer, thank you for your constructive questions: Our sample size remained consistent, with 1106 being the calculated sample size and 927 being the final achieved sample size.

On line 64, what is hospital malnutrition? That means patients developed undernutrition due to their hospitalization? It is better if replaced by malnutrition among adult patients admitted to the hospitals.

Response: Dear Reviewer, thank you for your constructive comments. We have made the modifications as recommended.

4. I am not satisfied with the conclusion section! Only you rewrote your results in the conclusion section. It is better if you add the consequence or impact of undernutrition.

Response: Dear Reviewer, thank you for your constructive comments and recommendations. We have revised the manuscript accordingly, and the changes can be found in the revised version of the manuscript.

5. On line 90, the first sentence of the paragraph needs a citation.

Response: Dear reviewer, thank you for your constructive recommendations. It was sited.

6.Line 110, regarding the gap of the study, I am not satisfied. Studies that were conducted in Ethiopia have limitations? In terms of variable or study design? Or did you because they are few in number? What were their gaps?

Response: Dear Reviewer, thank you for your constructive questions. Studies conducted in Ethiopia on adult malnutrition in hospital settings are limited in number and scope. Furthermore, malnutrition among adult hospitalized patients is not routinely recognized as a major clinical problem in many hospital settings, which highlights the importance of the present study.

7.line 125, you stated, “According to the Oromia Regional Health Office, Zonal Health Office, and Health Information Management System of each hospital, these hospitals provide health care services for more than 3.1 million people in the East Wollega zone, 1.5 million people in West Shewa, and 1.2 million people in the Buno Bedele zone.” Do you mean that each hospital has its own separate catchment population? Is it possible? Please clarify!

Response: Dear reviewer, thank you for your constructive questions. It is not that each hospital has its own separate catchment population, but rather that the catchment populations mentioned are served by these hospitals.

8. Line 130, gynecology ward: was it included or excluded?

Response: Dear reviewer, thank you for your constructive feedback. Yes, it is.

9.line 132 you stated, “…..However, patients with physical disabilities such as kyphosis, scoliosis, and limb defects including deformities, as well as pregnant patients, unconscious or comatose patients, obstetrics patients, surgical emergency patients, and patients to be discharged within 72 hours were excluded from the study.” Is there a difference between pregnant and obstetric patients? Why did you mention obstetrics patients separately?

Response: Dear Reviewer, thank you for pointing out the typographical error. We have made the necessary corrections in the manuscript.

10.If you excluded patients who discharged within 3 (72 hrs.) days, how could you get this sample size? What was the hospital’s average length of hospital stay?

Response: Dear reviewer, thank you for your questions. The average length of stay was 6 days.

11.In lines 154, 155, and 156, the assessment was made by whom? Nurse or physician, but you didn't include physician in the data collection procedure. So justify your data quality?

Response: Dear reviewer, thank you for your constructive comments and suggestions. Each sub section of the medical history and physical examination for SGA has a detailed check list that both physicians and BSc nurses are familiar with. Additionally, the data collectors were clinically experienced individuals who received training on the tool and data collection methods.

12. Have you excluded patients who have no lower extremities (amputated patients) in the orthopedics ward?

Response: Dear Reviewer, thank you for your question. No, patients with lower extremity amputations were not excluded from the study

13.Line 193 you have did pretest. But you didn’t state the hospital. At the same hospital?

Response: Dear Reviewer, thank you for your constructive comment and suggestions. The study was not conducted in the same hospital; it was carried out at Gimbi Hospital.

14. Line 301 Please add CI

Response: Dear reviewer, thank you for your

---

## [Decision Letter · Decision Letter 1]

9 Apr 2026

PONE-D-25-57131R1Nutritional status and associated factors of hospitalized adult patients in comprehensive specialized hospitals of Western EthiopiaPLOS One

Dear Dr. Moti,

Thank you for submitting your manuscript to PLOS ONE. After careful consideration, we feel that it has merit but does not fully meet PLOS ONE’s publication criteria as it currently stands. Therefore, we invite you to submit a revised version of the manuscript that addresses the points raised during the review process.

We look forward to receiving your revised manuscript.

Kind regards,

Nagasa Dida, MPH

Academic Editor

PLOS One

**Journal Requirements:**

**Additional Editor Comments:**

Please thoroughly revise the entire document for typographical and grammatical errors.Online 47 - Unless you have two or more distinct sample populations, use the term "sample" rather than "samples." Use the past tense throughout the report and for your objectives (Line 112), except within the conclusion and recommendations sections.Provide a reference for the information used in the "Study Area" sectionFor the methodological clarification, while data collection lasted five months, the caseload for proportional allocation was based on only three months. Please clarify how you determined the specific number of samples for each month, week, or day.Under the operational definition, please specify the exact diseases being studied, as not all conditions are directly relevant to your outcome variable (nutritional status).The non-response rate is significantly high; please provide a clear explanation or justification for this figureAs "semi-urban" is not an official category in the current Ethiopian census or administrative system, please clarify its definition and criteria within your study

Reviewers' comments:

Reviewer's Responses to Questions

**Comments to the Author**

1. If the authors have adequately addressed your comments raised in a previous round of review and you feel that this manuscript is now acceptable for publication, you may indicate that here to bypass the “Comments to the Author” section, enter your conflict of interest statement in the “Confidential to Editor” section, and submit your "Accept" recommendation.

Reviewer #2: All comments have been addressed

Reviewer #5: All comments have been addressed

Reviewer #6: (No Response)

Reviewer #8: (No Response)

2. Is the manuscript technically sound, and do the data support the conclusions?

Reviewer #2: Yes

Reviewer #5: Yes

Reviewer #6: Yes

Reviewer #8: Yes

3. Has the statistical analysis been performed appropriately and rigorously?

Reviewer #2: Yes

Reviewer #5: Yes

Reviewer #6: Yes

Reviewer #8: Yes

4. Have the authors made all data underlying the findings in their manuscript fully available?

Reviewer #2: Yes

Reviewer #5: Yes

Reviewer #6: Yes

Reviewer #8: Yes

5. Is the manuscript presented in an intelligible fashion and written in standard English?

Reviewer #2: No

Reviewer #5: Yes

Reviewer #6: Yes

Reviewer #8: Yes

6. Review Comments to the Author

**Reviewer #2:** Dear Author/s,

Thank you for conducting research on the nutritional status of hospitalized adult patients in comprehensive specialized hospitals of Western Ethiopia. It gives a good overview of the nutritional status of hospitalized adult patients in western Ethiopia.

To sum up, I think you may have revised and answered my comments and suggestions, but I feel there is still some work to do to mature the manuscript. Especially, there are still some grammatical and technical errors in the presentation. A further proof reading is still needed to standardize the language of presentation.

Reviewer #5: (No Response)

Reviewer #6: (No Response)

Reviewer #8: (No Response)

7. PLOS authors have the option to publish the peer review history of their article (what does this mean?). If published, this will include your full peer review and any attached files.

Reviewer #2: No

Reviewer #5: No

Reviewer #6: **Yes:** Bikila Regassa Feyisa

Reviewer #8: No

---

## [Author Response · Author response to Decision Letter 2]

17 Apr 2026

Dear Editor,

We have carefully addressed all essential points and comments raised by the editor and reviewers. Please find attached the revised document titled “Response to Reviewers,” which provides detailed responses to each comment.

Sincerely!

---

## [Editor Report · Decision Letter 2]

21 Apr 2026

Nutritional status and associated factors among hospitalized adult patients in comprehensive specialized Hospitals of Western Ethiopia

PONE-D-25-57131R2

Dear Mrs. %Moti%,

We’re pleased to inform you that your manuscript has been judged scientifically suitable for publication and will be formally accepted for publication once it meets all outstanding technical requirements.

Kind regards,

Nagasa Dida, MPH

Academic Editor

PLOS One

---

## [Editor Report · Acceptance letter]

PONE-D-25-57131R2

PLOS One

Dear Dr. Moti,

I'm pleased to inform you that your manuscript has been deemed suitable for publication in PLOS One. Congratulations! Your manuscript is now being handed over to our production team.

Kind regards,

on behalf of

Mr. Nagasa Dida

Academic Editor

PLOS One